# Endothelial-Specific Molecule 1 Inhibition Lessens Productive Angiogenesis and Tumor Metastasis to Overcome Bevacizumab Resistance

**DOI:** 10.3390/cancers14225681

**Published:** 2022-11-18

**Authors:** Nannan Kang, Xue Liang, Buxi Fan, Chen Zhao, Beiyu Shen, Xuemei Ji, Yu Liu

**Affiliations:** 1School of Life Science & Technology, China Pharmaceutical University, Nanjing 211198, China; 2Nanjing University of Chinese Medicine, Nanjing 210029, China

**Keywords:** endothelial cell specific molecule-1 (ESM1), tumor angiogenesis, bevacizumab resistance, delta-like ligand-4, anti-ESM1 monoclonal antibody

## Abstract

**Simple Summary:**

Bevacizumab mediated anti-angiogenesis provides a new chance of survival for patients with a tumor. However, most patients have acquired bevacizumab resistance after continuous treatment, and there is no available clinical therapy to overcome drug resistance. In this study, we analyzed unique expression patterns of genes from bevacizumab-sensitive or acquired bevacizumab-resistant cancer cells using RNA-seq analysis to identify the potential molecules and mechanism of bevacizumab resistance. We found that endothelial-specific molecule 1 expression elevated bevacizumab-resistant tumor cells, and endothelial-specific molecule 1 further regulates MMP9, VEGF, and DLL4 to promote metastasis and angiogenesis in vitro and in vivo. The anti-ESM1 monoclonal antibody developed by us significantly strengthened the efficacy of bevacizumab in vivo. This research has an important theoretical and clinical application value for elucidating the resistance mechanism and overcoming the bevacizumab resistance.

**Abstract:**

The development of drug resistance in malignant tumors leads to disease progression, creating a bottleneck in treatment. Bevacizumab is widely used clinically, and acts by inhibiting angiogenesis to “starve” tumors. Continuous treatment can readily induce rebound proliferation of tumor blood vessels, leading to drug resistance. Previously, we found that the fragment crystallizable (Fc) region of bevacizumab cooperates with the Toll-like receptor-4 (TLR4) ligand to induce M2b polarization in macrophages and secrete tumor necrosis factor-α (TNFα), which promotes immunosuppression, tumor metastasis, and angiogenesis. However, the downstream mechanism underlying TNFα-mediated bevacizumab resistance requires further investigation. Our RNA-Seq analysis results revealed that the expression of endothelial cell specific molecule-1 (ESM1) increased significantly in drug-resistant tumors and promoted metastasis and angiogenesis in vitro and in vivo. Furthermore, TNFα induced the upregulation of ESM1, which promotes metastasis and angiogenesis and regulates matrix metalloprotease-9 (MMP9), vascular endothelial growth factor (VEGF), and delta-like ligand-4 molecules (DLL4). Accordingly, the curative effect of bevacizumab improved by neutralizing ESM1 with high-affinity anti-ESM1 monoclonal antibody 1-2B7 in bevacizumab-resistant mice. This study provides important insights regarding the molecular mechanism by which TNFα-induced ESM1 expression promotes angiogenesis, which is significant for elucidating the mechanism of bevacizumab drug resistance and possibly identifying appropriate biosimilar molecules.

## 1. Introduction

The outgrowth of blood vessels to form neovasculature is a fundamental process in solid tumor progression as tumors larger than 2 mm in diameter depend on blood vessels for nutrients and oxygen. Moreover, myriad angiogenic factors are secreted by tumors to induce angiogenesis and establish a nutrient metabolism network for tumor growth. Vascular endothelial growth factor (VEGF) and its receptor (VEGFR) constitute the primary activators of a critical pro-angiogenic signaling pathway. This pathway has attracted vast attention as a possible target for anti-tumor strategies, including inhibiting VEGF expression, blocking tumor cell signal transduction, and exhausting VEGF produced by tumor cells. Bevacizumab was the first humanized monoclonal antibody to inhibit neovascularization by binding soluble VEGF A and, subsequently, blocking the VEGFA/VEGFR interaction. Bevacizumab has been approved by the Food and Drug Administration (FDA) and European Medicines Agency (EMA) as an anti-tumor agent [1]. Neoadjuvant therapy with bevacizumab combined with chemotherapy has been applied to various types of solid tumors, including non-small cell lung cancer [2], breast cancer [3], glioblastoma [4], and ovarian cancer [5]. However, the main challenge facing anti-VEGF therapy is drug resistance, which significantly limits long-term treatment [6].

Increasing evidence indicates that activation of alternative proangiogenic factors, such as b-FGF, PIGF, HGF, EGF, VEGF C, and angiopoietin [7,8], is associated with bevacizumab resistance. Furthermore, several mechanisms of bevacizumab resistance have been reported in glioblastoma [7,8,9,10], ovarian cancer [11], and non-small cell lung cancer [12]. These mechanisms include: activation of the alternative pathway; (2) increased proliferation, migration, and invasion of tumors [13]; vascular disorders caused by tumor hypoxia [14] and tumor metastasis [12]; recruitment of vascular progenitor cells and regulatory factors [15]; promotion of myeloid-derived suppressor cells, tumor-associated macrophages, cancer-associated fibroblasts, and non-cellular components, such as the extracellular matrix and cytokines in the microenvironment [10,16,17]; and production of vesicles containing proangiogenic molecules, such as VEGF, matrix metalloprotease-9 (MMP9), and hypoxia-induced factor-1α (HIF1α) [18,19]. Among these potential targets, angiogenin-2 (Ang-2) appears to be the most promising candidate for overcoming bevacizumab resistance. An Ang-2/VEGF bi-specific antibody (vanucizumab) was developed by Genentech (South San Francisco, CA, USA); however, the clinical trials were terminated before phase III owing to insufficient efficacy compared to bevacizumab alone. Therefore, intensive characterization of the molecular mechanisms that mediate anti-angiogenesis resistance may provide potential strategies to improve therapeutic efficacy and prolong patient prognosis.

Endothelial cell-specific molecule-1 (ESM1) is a secreted proteoglycan comprising a 20 kDa protein core and a dermatan sulfate [20], that was first cloned from HUVEC cells [21]. ESM1 is highly expressed in vascular endothelial cells, i.e., epithelial cells of the distal renal tubules, bronchial tubes, and submucosal lung glands in normal tissues [22]. ESM1 expression is also upregulated in lung cancer [23], gastric cancer [24], breast cancer [25], bladder cancer [26], and other malignant tumors and is associated with the inflammatory response and tumor progression. Moreover, ESM1 participates in tumor growth, migration, invasion, and angiogenesis [27]. Scherpereel et al. found that tumors did not develop via subcutaneous injection of normal HEK293 cells but were formed by ESM1-overexpressing HEK293 cells in mice [28]. Additionally, inhibiting ESM1 mRNA with siRNA reduces MMPs and epithelial-mesenchymal transition (EMT)-related gene expression, thus inhibiting tumor invasion in a colorectal cancer model [29]. ESM1 is also highly expressed in “tip cells”, which mediate vascular growth, highlighting the potential role of ESM1 in vascular network formation [30]. However, direct evidence to demonstrate the role of ESM1 in tumor angiogenesis, particularly in anti-angiogenesis therapy resistance, requires further investigation.

Angiogenesis requires multifaceted adjustment and balance during embryonic development and tumor growth. Gene-targeting approaches have confirmed that vascular system development is impaired without VEGF in embryonic mice, suggesting a fundamental role for VEGF in embryonic development and tumor angiogenesis [31]. The Delta/Jagged–Notch system also regulates cell fate, thus influencing cell proliferation, differentiation, and apoptosis. Delta-like ligand-4 (DLL4) is a membrane ligand of Notch1 and is required for normal vascular development and arterial formation in mice [32,33]. Heterozygous DLL4 mutations can be lethal to embryos by causing a lack of well-defined major arteries and an increased number of vessel branches and vascular sprouts [32,33]. Moreover, DLL4 is strongly expressed in renal cell carcinoma, stomach cancer, colorectal cancer, and metastatic breast cancer [34,35,36]. A high level of DLL4 is associated with a reduced efficacy of bevacizumab in advanced colorectal cancer [37]. DLL4 is normally induced by VEGF and is a negative-loop feedback regulator that inhibits vascular sprouting and branching [31]. VEGF blockade attenuates the formation of new tumor vessels, and normalization of the remnant vessels leads to tumor recession. Similarly, DLL4 blockade decreases tumor growth as DLL4 inhibition increases the density of poorly functional vascular sprouts and branches, inducing disordered tumor vessels. In humans, VEGF and DLL4 are strongly co-expressed in tumor tissues [38]. Thus, the DLL4/Notch pathway is being explored as an alternative target for antiangiogenic therapies, and several bispecific antibodies target DLL4 and VEGF, including dilpacimab [39] and ABL001 [38]. However, the molecular mechanism underlying the cooperation of DLL4 and VEGF to promote angiogenesis needs to be further studied.

Our group previously reported that the curative effect of bevacizumab is significantly improved by neutralizing M2b macrophage-related TNFα with an anti-TNFα nanobody [16]. In this study, we aimed to find the potential downstream molecules of TNFα-induced resistance to bevacizumab. The unique expression patterns of genes from bevacizumab-sensitive or acquired bevacizumab-resistant cancer cells was analyzed using RNA-seq analysis. We proved that expression of ESM1 is regulated by TNFα and contributes to acquired bevacizumab resistance. Finally, we hypothesized that the inhibition or deletion of ESM1 might reverse bevacizumab resistance and verified our hypothesis. This study is essential to understand how bevacizumab resistance develops and how it may be overcome.

## 2. Materials and Methods

### 2.1. Animal Experiments

Female BALB/c nude mice (6–8 weeks old) were purchased from the Model Animal Research Center of Nanjing University (Nanjing, China) and housed according to the guidelines of the Animal Care and Use Committee of China Pharmaceutical University (Nanjing, Jiangsu, China). The mice were injected subcutaneously in the left flank with 1 × 10^7^ MDA-MB-231-S, MDA-MB-231-S^ovESM1^, MDA-MB-231-R, or MDA-MB-231-R-ESM1^ko^ cells, or 3 mm^3^ tumor tissue (*n* = 5–7). On day 7–15 following tumor cell/tissue injection, they were intraperitoneally administered with bevacizumab and/or anti-ESM1 monoclonal antibody 1-2B7 (10 mg/kg i.p. for bevacizumab, 10 mg/kg i.p., and 20 mg/kg i.p. once per week), sorafenib (30 mg/kg i.p., once per day), docetaxel (10 mg/kg i.p., every 4 days), or phosphate-buffered saline (PBS). After 2–3 weeks of treatment, all mice were sacrificed. Mice were euthanized through cervical dislocation. The specific operations are: Mice were first anesthetized. Then, researchers caught the tail root and lifted it on the cover of the mice or other rough surface with the right hand, used the thumb and index finger of the left hand to press down the head and neck of the mice, and grabbed the root of the tail of the mice with the right hand to pull back and up which resulted in cervical dislocation. The spinal cord and brain stem were severed, and the experimental animal died immediately. The toxicity parameter following the injection is the weights of the mice. We ended our experiments when the weight of any group of balb/c-nude mice was lower than 17 g. The tumor volume was calculated using the formula V = 0.52 × A × B × B, where V represents the tumor volume, A is the longitudinal length of the tumor, and B is the transverse width of the tumor. The mouse weight and tumor volume were assessed every 3 days.

### 2.2. Cell Culture

Human breast cancer cell line MDA-MB-231 was purchased from the American Type Culture Collection (ATCC, USA) and was identified by cell STR identification. Cells were maintained at 37 °C and 5% CO_2_ in DMEM supplemented with 10% FBS (fetal bovine serum).

### 2.3. Real-Time Polymerase Chain Reaction (PCR) Analysis

Total RNA was extracted and reverse transcribed with HiScript II Q Select RT Super Mix for qPCR (Vazyme Biotech, Nanjing, China). Quantitative real-time PCR (qRT-PCR) was performed using CFX96™ and CFX384™ Real-Time PCR Detection Systems (Bio-Rad, Hercules, CA, USA). Relative quantification of mRNA levels was conducted using the comparative Ct method with Gapdh as the reference gene and the formula 2^−ΔΔCt^. Primers used for qRT-PCR were downloaded from Primer Bank (https://pga.mgh.harvard.edu/primerbank/) accessed on 14 September 2021.

### 2.4. ESM1 Knockdown or Knock-Out Assay

293T cells were infected with pCMV-VSVG, psPAX2, and pLKO.1 lentiviral plasmids expressing non-targeting shRNA control or human ESM1 shRNAs (Genscript, Nanjing, China) or pCMV-VSVG, psPAX2, and lentiCRISPR v2 lentiviral plasmids expressing human ESM1 sgRNAs. After 24 or 48 h, culture supernatant was collected and added to MDA-MB-231-S cells in the presence of 4 µg/mL polybrene. After 48 h, subcultured cells were selected in 1 µg/mL puromycin for 1 week. Lysates from stably selected cells were assessed for ESM1 expression using Western blotting or dot blotting.

### 2.5. Immunohistochemistry Staining

Tissue sections were incubated overnight with primary antibodies against VEGFA (19003-1-AP; Proteintech, USA), DLL4 (21584-1-AP; Proteintech), CD31 (GB11063-1; Servicebio, China), α-SMA (ab7817; Abcam, Cambridge, UK), and ESM1 (ab103590; Abcam, Cambridge, UK). They were subsequently incubated with goat anti-rabbit IgG H&L (ab150077; Abcam, Cambridge, UK), goat anti-mouse IgG H&L (ab150117; Abcam, Cambridge, UK), or goat anti-rat IgG H&L (ab150167; Abcam, Cambridge, UK) secondary antibodies. The slides were washed and treated with diaminobenzidine chromogen for 3 to 5 min, yielding a dark brown color. The sections were counterstained with hematoxylin and scanned at ×40 using a Nano Zoomer 2.0 HT (Hamamatsu Photonics K. K., Hamamatsu, Japan).

### 2.6. Cell Migration and Invasion

MDA-MB-231-S/E/R or ESM1 knock-down cell invasion were evaluated using a Transwell chamber (Corning, New York, NY, USA). Cells (1 × 10^5^ cells/mL) were plated on DMEM-diluted matrigel (50 μL/cm^2^) in the upper chamber, and DMEM medium (added 10% FBS) was added to the lower chamber. After 24 h of incubation at 37 °C, the cells were fixed in 4% paraformaldehyde and stained with crystal violet dye; the migrated cells were in the lower surface of the filter. Results were obtained by averaging the total number of cells from three fields of views.

### 2.7. MTT Assay

HUVEC cells (5000 cells/well) or MDA-MB-231-S/E/R cells (5000 or 10,000 cells/well) were seeded in 96-well plates (Thermo Fisher, Waltham, MA, USA), and the media were replaced with conditioned medium from MDA-MB-231-S/E/R or ESM1 knock-down cells (for HUVEC cells) or fresh medium with 10% FBS (for MDA-MB-231-S/E/R cells) in 12 h later. After 12–36 h, 20μL of MTT (5 mg/mL) were added into each well and incubated at 37 °C for 4 h. Then, the media in each well were removed and 100μL of DMSO was added into each well. Finally, A490 nm was measured using a Multiskan™ FC microplate reader (Thermo Fisher, Waltham, MA, USA).

### 2.8. ELISA

Serum and cellular supernatant VEGFA were measured using enzyme linked immunosorbent assay (ELISA) with a Human VEGFA ELISA Kit (CHE0043, 4A Biotech, Beijing, China), according to the manufacturer’s instructions. The cellular supernatant samples were diluted twice, and the results were expressed as pg/mL.

### 2.9. Anti-ESM1 mAbs Preparation

Female BALB/c mice (6–8 weeks old) were immunized with hESM1 mixed with complete Freund’s adjuvant (Sigma Aldrich, Darmstadt, Germany) and incomplete Freund’s adjuvant (Sigma Aldrich, Darmstadt, Germany) and fused with mouse myeloma cells SP2/0 using the hybridoma technique. Eleven monoclonal antibodies against human ESM1 were obtained after subclonal screening and ELISA identification.

### 2.10. Statistical Analyses

The data are presented as mean ± standard deviation (SD) using GraphPad Prism 7.0. All the experiments were repeated independently at least thrice. *p* values < 0.05 were considered statistically significant.

## 3. Results

### 3.1. Bevacizumab-Resistant Tumor Cells Become More Aggressive In Vitro and In Vivo

In our previous study, we successfully developed an animal model (bev-resistance-balb/c-nu) based on continuous drug administration to MDA-MB-231-bearing mice to acquire bevacizumab resistance. Microvessel formation and tumor growth advanced in the bev-resistance-balb/c-nu model compared with that in PBS-treated mice despite continuous bevacizumab treatment [16]. To explore whether the bevacizumab-resistant model was consistent with explosive tumor proliferation and rapid metastasis in clinical patients following development of bevacizumab resistance, we isolated bevacizumab-resistant tumor cells from bev-resistance-balb/c-nu mice, named MDA-MB-231-R (resistant), and assessed their proliferation and invasiveness relative to that of MDA-MB-231-S (sensitive) cells. We seeded 5000 or 10,000 cells into 96-well plates, cultured them for 48 h, and measured the absorbance at 490 nm following addition of 3-(4,5-Dimethylthiazol-2-yl)-2,5-diphenyltetrazolium bromide (MTT) for 4 h. The proliferation capacity of MDA-MB-231-R cells was significantly higher that of MDA-MB-231-S (Figure 1a). Similar results were obtained in the Matrigel invasion assay: MDA-MB-231-R cells had a greater ability to invade under the same conditions (Figure 1b). Subsequently, we injected MDA-MB-231-R or MDA-MB-231-S cells into one flank of mice (*n* = 15/group), along with bevacizumab administration. Tumors volumes of all mice were measured, and tumor samples and serum were randomly derived from three mice every 3 days. The tumors progressed rapidly in MDA-MB-231-R-treated mice, indicating that bevacizumab therapy was ineffective compared to that in the MDA-MB-231-S group (Figure 1c). Similarly, microvessel development was more advanced in the MDA-MB-231-R group compared with that in the MDA-MB-231-S group (Figure 1d). To determine whether there is a potential difference in metastatic capacity of MDA-MB-231-S/R treated with bevacizumab in vivo, we co-stained tumors for E-cadherin and vimentin to detect tumor EMT. Vimentin level was markedly higher in MDA-MB-231-R tumors than in MDA-MB-231-S cells (Figure 1e). To quantify lung metastases, MDA-MB-231-R and MDA-MB-231-S cells were injected into mice through the tail vein, and lung histological sections were observed at 2 and 4 weeks. Alien cells infiltrated the lungs and formed nodules in the MDA-MB-231-R group (Figure 1f). Therefore, bevacizumab-resistant MDA-MB-231 cells have increased proliferation and invasion capacity in vitro and the ability to colonize the lungs in vivo.

### 3.2. ESM1 Is Overexpressed in Bevacizumab-Resistant Tumor Cells and Correlates with High Occurrence of Metastasis

To determine why bevacizumab-resistant cells were more aggressive than bevacizumab-sensitive cells, RNA-seq analysis was performed for MDA-MB-231-S and MDA-MB-231-R cells. A total of 305 genes were upregulated and 886 were downregulated in MDA-MB-231-R cells compared to MDA-MB-231-S cells. Among them, ESM1 was strongly upregulated and simultaneously correlated with angiogenesis-related VEGFA (R = 0.48), ANGPT2 (R = 0.66) and M2b macrophage related cytokines, such IL6, TNFα, and IL10, analyzed using the String database (https://www.string-db.org/) accessed on 6 March 2022 (Figure 2a–c). Moreover, it is reported that inflammatory cytokines, such as TNFα and IL-1β, can stimulate ESM1 secretion [40]. Therefore, we hypothesized that ESM1 may be a downstream molecule of TNFα that mediates bevacizumab resistance and further verified whether ESM1 was highly expressed in bevacizumab-resistant tumor tissues or cells. qPCR results revealed that ESM1 was increased by approximately 9.21− and 2.06-fold in bevacizumab-resistant tumor tissues and cells, respectively, compared to controls (Figure 2d,e). Given that ESM1 is a secretory proteoglycan, we quantified the protein levels in bevacizumab-resistant and bevacizumab-sensitive cells using dot blot (Figure 2f) and found that ESM1 expression was higher in bevacizumab-resistant tumor cells. Moreover, ESM1 seems upregulated in many tumor tissues than normal tissues in The Cancer Genome Atlas (TCGA) database (Figure 2g).

To determine whether ESM1 causes bevacizumab-resistant tumor metastasis, we overexpressed ESM1 in MDA-MB-231-S cells and named them MDA-MB-231-E (MDA-MB-231-S ^ovESM1^; Figure 3a). ESM1 significantly increased the invasiveness of MDA-MB-231-S cells (Figure 3b). Moreover, in a in balb/c-nu subcutaneous transplantation tumor model, the tumor burden of mice receiving MDA-MB-231-E cells was significantly larger than that of MDA-MB-231-S cells treated with PBS or bevacizumab (Figure 3c). Since both MDA-MB-231-S and MDA-MB-231-E were labeled with enhanced green fluorescent protein (EGFP), we also analyzed the micro-metastasis and metastatic nodules in mouse lungs. Elevated EGFP-labeled cells infiltrated the lungs of MDA-MB-231-E subcutaneous transplantation tumor-bearing mice treated with PBS or bevacizumab (Figure 3d), and metastatic nodules were observed (Figure 3e). Additionally, more microvessels were observed in the MDA-MB-231-E than MDA-MB-231-S group after staining with CD31 (Figure 3f). Interestingly, we found that bevacizumab administration further increased ESM1 and CD31 expression in tumors in vivo (Figure 3g), indicating that bevacizumab resistance and ESM1 expression are reinforced mutually. To further confirm that ESM1 could promote tumor lung metastasis, we injected 5 × 10^5^ MDA-MB-231-S, MDA-MB-231-E, MDA-MB-231-R or shESM1-MDA-MB-231-R cells into balb/c-nu mice via the tail vein. After 4 weeks, hematoxylin and eosin (H&E) staining was performed and showed that ESM1 contributed to tumor cell colonization of the lung (Figure 3h). Moreover, we tested whether increased metastasis was due to up-regulation of MMPs by ESM1 overexpression, as it was reported that ESM1 up-regulated MMP9 leading to radiotherapy resistance [25]. MMP9 mRNA and protein expression was significantly higher in MDA-MB-231-E and MDA-MB-231-R cells relative to MDA-MB-231-S (Figure 3i,j). However, MMP9 expression and lung metastasis decreased when ESM1 was deleted by shRNA (Figure 3j,k). Collectively, these data suggest that ESM1 significantly enhanced tumor invasion in vitro and in vivo and accelerated tumor growth in vivo.

### 3.3. ESM1 Is Regulated by TNFα- NF-κB- RelB Axis

To ascertain why ESM1 is upregulated in bevacizumab-resistant tumor cells, we evaluated the effects of different concentrations of TNFα on ESM1 mRNA and protein expression in MDA-MB-231-S cells. We previously reported that TNFα was a key bevacizumab resistance-promoting effector molecule secreted by M2b macrophages [16]. Moreover, inflammatory cytokines, such as TNFα and IL-1β, can stimulate ESM1 secretion [40], but the molecular mechanism has not been elucidated. Following exposure of MDA-MB-231-S cells to different doses of TNFα for 24 h, the expression of ESM1 was enhanced by TNFα in a dose-dependent manner (Figure 4a,b). Given that NF-κB is a downstream signaling pathway of TNFR [41] and commonly regulates gene expression via a subunit of NF-κB, especially RelA (p65) or the RelB, we postulated that TNFα promotes ESM1 expression via a subunit of NF-κB. Hence, we overexpressed RelA or RelB in MDA-MB-231-S cells (Figure 4c) and found that ESM1 increased only upon the overexpression of RelB, not RelA (Figure 4d). To investigate whether RelA or RelB regulates ESM1 by affecting promoter activation of ESM1, we constructed a luciferase reporter plasmid (pGL3-basic) containing the ESM1 promoter region from −2000 to 0 (Pwt-Luc; Figure 4e). Overexpression of RelB significantly enhanced Pwt-Luc-driven luciferase activity in HEK293T cells (Figure 4f). Furthermore, we found two potential RelB-target sites in the promoter of ESM1 located in the −1484 to −1474 and −436 to −426 regions using the JASPAR database (http://jaspar.genereg.net/) accessed on 7 July 2021. We then constructed two mutant reporter plasmids with these two regions (Pmut1-Luc and Pmut2-Luc, respectively; Figure 4e). Overexpression of RelB significantly increased Pwt-Luc and Pmut1-Luc, but not Pmut2-Luc-driven luciferase, in HEK293T cells (Figure 4g), indicating that the −436 to −426 region of the ESM1 promoter is the RelB binding site. Collectively, these results indicated that TNFα stimulate ESM1 expression via RelB regulating the AAGGAGAATTA sequence at the ESM1 promoter to activate its transcription.

### 3.4. ESM1 Promoted the Proliferation and Migration of Endothelial Cell In Vitro

Given that significant up-regulation of CD31 was observed in tumors overexpressing ESM1 in vivo (Figure 3e and Figure 5a), we also sought to decipher the role of ESM1 in angiogenesis. The first step in the formation of new blood vessels was the proliferation and migration of vascular endothelial cells [42]. HUVECs were incubated with conditioned media from MDA-MB-231-S/E/R or shESM1-MDA-MB-231-R cells. HUVEC proliferation was found to be higher in the conditioned medium from MDA-MB-231-E/R compared to that in MDA-MB-231-S; ESM1 knockdown in MDA-MB-231-R cells reduced HUVEC proliferation (Figure 5b–d). We then performed a wound healing assay to confirm the role of ESM1 in HUVEC migration. The supernatant of MDA-MB-231-E or MDA-MB-231-R cell lines, which highly express ESM1, induced HUVEC migration compared with MDA-MB-231-S cells; ESM1 knockdown in MDA-MB-231-R cells reduced HUVEC migration after 24 or 36 h (Figure 5e,f). Collectively, these data suggested that ESM1 promoted endothelial cell behaviors associated with angiogenesis in vitro.

### 3.5. Effect of ESM1 on In Vivo Productive Angiogenesis, VEGF and DLL4

A previous study reported that ESM1 played a fundamental role by promoting VEGF binding to VEGFR2 [43]. Therefore, we investigated whether ESM1 affects the expression and the function of VEGF. To this end, we quantified VEGF in conditioned medium from MDA-MB-231-S/E cells via ELISA. MDA-MB-231-E cells secreted more VEGF than MDA-MB-231-S cells (Figure 5g,h). Furthermore, ESM1-overexpressing HUVECs had higher VEGF mRNA and protein expression levels (Figure 5i,j). We further analyzed serum VEGF levels of MDA-MB-231-R/E/S tumor-bearing mice treated with bevacizumab, and found them to be higher in the MDA-MB-231-E and MDA-MB-231-R groups compared to the MDA-MB-231-S group (Figure 5k,l). Furthermore, to ascertain the functional role of VEGF in the observed ESM1-induced endothelial cell proliferation and migration, we added bevacizumab (10 μg/mL) to the conditioned medium from MDA-MB-231-S/E/R cells as a neutralizing antibody to block VEGF and found that the proliferation of HUVEC was reversed (Figure 5m), while migration was not after 12 h (Figure 5e,n). Together, these results indicate that ESM1 promoted HUVEC proliferation by upregulating VEGF expression. The migration of HUVEC seems not to have been influenced powerfully by VEGF-blocking in the conditioned medium.

It is reported that not only VEGFA-VEGFR but also the DLL4/Notch signaling pathway is involved in vascular proliferation and blood vessel growth [44]. Moreover, in human breast cancer, ESM1 was correlated with DLL4 in the GEPIA database (R = 0.34, *p* = 0; Figure 6a,b) and DLL4 was increased in ESM1-treated HUVEC cells (Figure 6c). In human colorectal cancer, inhibition of the Notch/DLL4/Hes pathway significantly inhibited angiogenesis [45]. We further evaluated the expression of DLL4 in ESM1-normal, -high, or -low expression tumor cells and found that DLL4 expression was higher in MDA-MB-231-R/E cells compared to MDA-MB-231-S and shESM1-MDA-MB-231-R cells (Figure 6d,e). Hence, ESM1 not only upregulated VEGF but also DLL4 expression. Considering that DLL4 expression in BRCA patients was moderately correlated with the expression of VEGF in the TCGA database (R = 0.18, *p* < 0.0001; Figure 6f), we next examined the correlation between VEGF/ESM1/DLL4 expression in vivo and found that VEGF and DLL4 were significantly upregulated in ESM1-overexpressing MDA-MB-231 cell-derived tumors (Figure 6g). In addition, the vasculature in MDA-MB-231-E cells bearing mice-derived tumors, which have more overlap regions of CD31 and α-SMA, appeared to be more integrated than that in MDA-MB-231-S cells (Figure 6h). Collectively, these findings suggest that ESM1 overexpression promoted productive angiogenesis. Furthermore, ESM1 upregulated two angiogenesis related ligands, VEGF and DLL4.

### 3.6. ESM1 Deletion in Bevacizumab Resistant Cells Inhibits Tumor Growth In Vivo

To better investigate the biological function of ESM1 in bevacizumab resistance in vivo, ESM1 was effectively deleted using the Crispr-Cas9 system (Figure 7a). MDA-MB-231-R or MDA-MB-231-R ESM1^ko^ cells (10^7^) were implanted subcutaneously into the flanks of balb/c-nude mice. When tumors reached a volume of 100 mm^3^, bevacizumab was administered once per week (Figure 7b). Compared to MDA-MB-231-R-bearing mice, tumor growth was remarkably suppressed in mice with MDA-MB-231-R ESM1^ko^ treated with bevacizumab (Figure 7c,d). Furthermore, ESM1 deletion reduced CD31 levels in xenografts and was involved in decreased tumor angiogenesis. Moreover, ESM1 deletion led to more non-productive blood vessels that appeared disordered and disconnected, indicating a lack of nutritional transport (Figure 7e). ESM1 deletion caused significant necrosis in the tumor, which may have been due to the impaired nutritional transport function of microvessels (Figure 7f). Additionally, ESM1 deletion reduced EMT and metastasis to the lungs (Figure 7g,h). These findings strongly suggest that ESM1 deletion enhanced bevacizumab efficacy and reduced functional tumor angiogenesis and lung metastasis in vivo.

### 3.7. Targeting ESM1 with an Anti-ESM1 Monoclonal Antibody on Bevacizumab Resistance Increased the Effect of Bevacizumab

Finally, we prepared anti-ESM1 monoclonal antibodies and selected the 1-2B7 antibody, with high affinity (1.37 nM) as an ESM1 neutralizing antibody. We treated bevacizumab-resistant tumor-bearing mice with bevacizumab, 1-2B7, bevacizumab and 1-2B7, sorafenib, docetaxel or normal saline (Figure 8a). Compared to those treated with bevacizumab alone, tumors treated with both bevacizumab and 1-2B7 exhibited significant tumor growth suppression (Figure 8b–d). Tumor necrosis in mice receiving bevacizumab combined with ESM1-neutralizing antibodies was significantly higher compared to that in the bevacizumab only group (Figure 8e). Moreover, bevacizumab combined with 1-2B7 reduced tumor angiogenesis as well as the quantity of functional blood vessels (Lectin+) (Figure 8f). In summary, these data underscore the effectiveness of the anti-ESM1 monoclonal antibody 1-2B7 in vivo, further strengthening our hypothesis that blocking ESM1, when combined with bevacizumab administration, enhances the therapeutic effects of anti-angiogenesis therapy.

## 4. Discussion

Angiogenic factors are important for tumor vasculature development and homeostasis. ESM1 is responsible for modulating growth factors, chemokines, inflammation, and angiogenesis [46]. Here, we found that ESM1 was overexpressed in bevacizumab-resistant triple-negative breast cancer cells and was involved in increased tumor angiogenesis and metastasis. We provided comprehensive evidence to show that ESM1 is upregulated by TNFα and further regulated VEGF and DLL4 to promote pathologically productive angiogenesis. Deletion or neutralization of ESM1 significantly inhibited tumor growth and overcame bevacizumab resistance in triple negative breast cancer, thus confirming that ESM1 was a key molecule mediating bevacizumab resistance.

Notably, the expression of ESM1 was approximately 9.2 times higher in drug-resistant tumor cells than in sensitive cells in vivo (Figure 2). However, after the bevacizumab-sensitive and bevacizumab-resistant cells were separated from tumor-bearing mice, ESM1 expression in drug-resistant cells was still stable and highly expressed, with 2.1 times that of sensitive cells. Continuous bevacizumab treatment promoted the polarization of M2b macrophages in the TME, resulting in TNFα-induced high ESM1 expression in tumor cells in vivo. However, the resistant tumor cells isolated from mice maintained high ESM1 expression under bevacizumab maintenance treatment, which may be due to the acquisition of ESM1 high expression gene mutation in resistant tumor or the existence of a compensatory mechanism to regulate ESM1 expression in vitro. Our study elucidated the molecular mechanism by which the TNFα-RelB-ESM1 signaling axis regulated ESM1 expression, and more mechanisms need to be further studied. Furthermore, the ESM1 monoclonal antibody alone was less effective than bevacizumab but slightly superior to the control group (Figure 8c–e). Retinal vascular outgrowth and filopodia emission were impaired in ESM1^KO^ mice [47], and overexpression of ESM1 induce tumor formation [28]. Our results show that ESM1 regulated VEGF expression; however, ESM1 is not the only molecule to regulate the VEGF in cells. In addition to VEGF, ESM1 promoted MMP9, DLL4, and ICAM3 expression, indicating that ESM1 cooperated with VEGF to promote tumor development and promote bevacizumab resistance. Moreover, given that ESM1 is a soluble proteoglycan, developing dual-target antibody drugs against ESM1 and VEGF may be a feasible strategy.

In addition, ESM1 is secreted primarily by endothelial cells in physiological homeostasis, and its content in normal body fluids is much lower than that in tumor patients; therefore, previous studies have focused on analyzing the relationship between ESM1 in body fluids and cancer prognosis [27,48,49]. Based on these data, ESM1 can be used as a prognostic or survival marker in patients. Our study revealed that tumor-secreted ESM1 played an important role in promoting anti-VEGF resistance. Therefore, nucleic acid drugs based on ESM1 silencing in tumor cells are a potential treatment strategy. We also found that docetaxel, a first-line drug for triple-negative breast cancer treatment, does not significantly impact the effect on bevacizumab alone after bevacizumab resistance. Whether bevacizumab resistance leads to tumor insensitivity to chemotherapy is a clinical question worth exploring.

## 5. Conclusions

This study elucidates why ESM1 is typically elevated in bevacizumab-resistant tumor cells and how it promotes bevacizumab resistance. Furthermore, high-affinity monoclonal antibody 1-2B7 against ESM1 combination with bevacizumab has a significantly improved therapeutic effect to overcome bevacizumab resistance. This study provides important insights for understanding the mechanism of bevacizumab resistance and proposes strategies to overcome this resistance in cancer therapy.

## Figures and Tables

**Figure 1 cancers-14-05681-f001:**
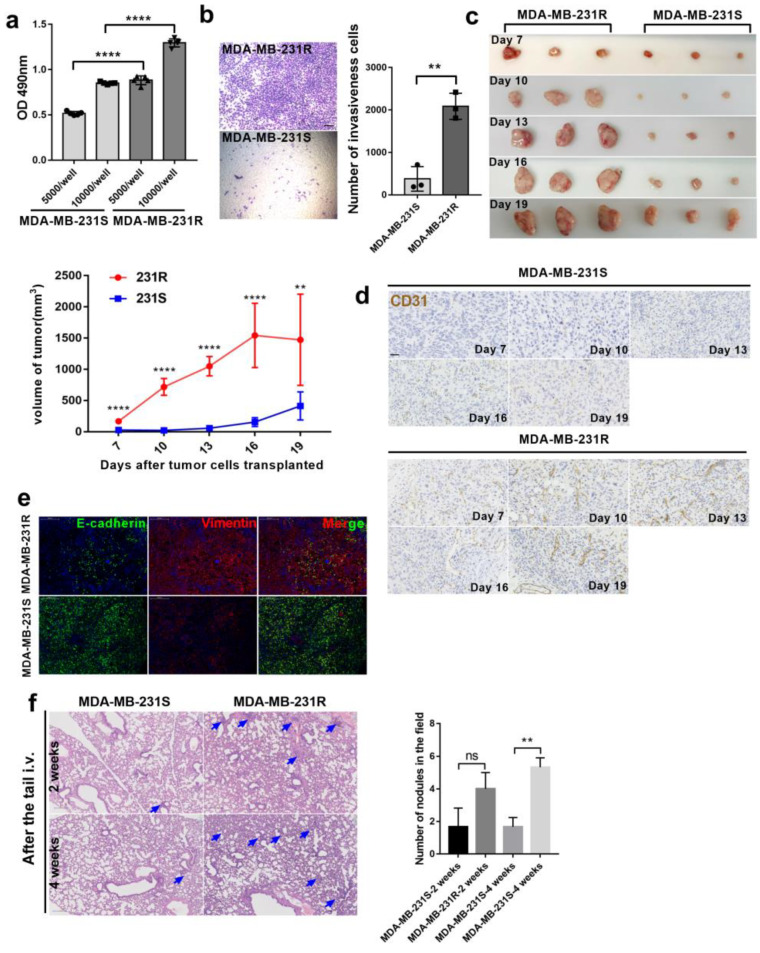
Evaluation of bevacizumab resistance model. (**a**) Different number of MDA-MB-231-S/R were seeded in 96-well-plate and cell survival was measured by MTT after 48 h. (MDA-MB-231-S, bevacizumab sensitive MDA-MB-231 cell; MDA-MB-231-R, bevacizumab resistant MDA-MB-231 cell), ****, *p* < 0.0001, *t*-test. Error bars represent SD. (**b**) The number of invaded cells in each group. *n* = 3. ** *p* < 0.01 by *t* test. Error bars represent SD; each transwell assay was independently repeated three times; scale bar, 100μm. (**c**) Tumor growth volume and photos over time following subcutaneous transplant of MDA-MB-231-S/R tumor tissues into nude-balb/c mice. *n* = 15 for each condition. ** *p* < 0.01; **** *p* < 0.0001 by two-way ANOVA test. (**d**) Representative images of tumor sections from two groups treated with bevacizumab at different time points, Scale bar, 200 μm. (**e**) Representative images of tumor sections from mice treated with bevacizumab. Vimentin was co-stained with E-cadherin and nucleus. Scale bar, 200 μm. (**f**) Representative images and statistical analysis of HE staining to assess the number of lung metastasis in the different treatment groups; scale bar, 100 μm, ** *p* < 0.01 by *t* test.

**Figure 2 cancers-14-05681-f002:**
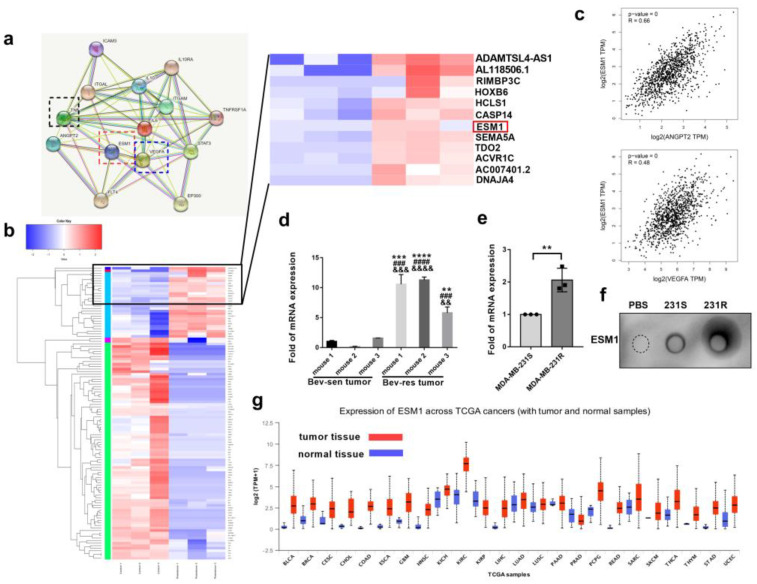
ESM1 is overexpressed in bevacizumab—resistance tumor tissues and cells. (**a**) Protein interaction network between ESM1 and VEGF/ANGPT2/M2b related cytokines in String Database. (Known interactions: light blue line, from curated databases; purple line, experimentally determined. Predicted interactions: green line, gene neighborhood; red line, gene fusions; blue line, gene co—occurrence. Others: yellow line, text mining; black line, co—expression; light purple line, protein homology.) (**b**) Expression pattern clustering heat map of MDA-MB-231-S and 231-R cells. (**c**) Gene correlation of ESM1 with VEGFA or ANGPT2 analyzed by GEPIA Database. (**d**) qRT—PCR of ESM1 mRNA levels in tumor tissues from bevacizumab sensitive and resistant mice. *n* = 3/group. ** *p* < 0.01; *** *p* < 0.001; **** *p* < 0.0001, by *t* test. Bev-res vs. mouse1 of Bev-sen group. ### *p* < 0.001; #### *p* < 0.0001, by *t* test. Bev-res vs. mouse 2 of Bev-sen group. && *p* < 0.01; &&& *p* < 0.001; &&&& *p* < 0.0001, by *t* test. Bev-resistance vs. mouse 3 of Bev-sensitive group. Error bars represent SD. (**e**) qRT-PCR of ESM1 mRNA levels in MDA-MB-231-S and MDA-MB-231-R. *n* = 3/group. ** *p* < 0.01, by *t* test. (**f**) Dot blotting of ESM1 in supernatants of PBS, MDA-MB-231-S, and MDA-MB-231-R under the same conditions. (**g**) ESM1 levels in tumor tissues and normal tissues in TCGA database.

**Figure 3 cancers-14-05681-f003:**
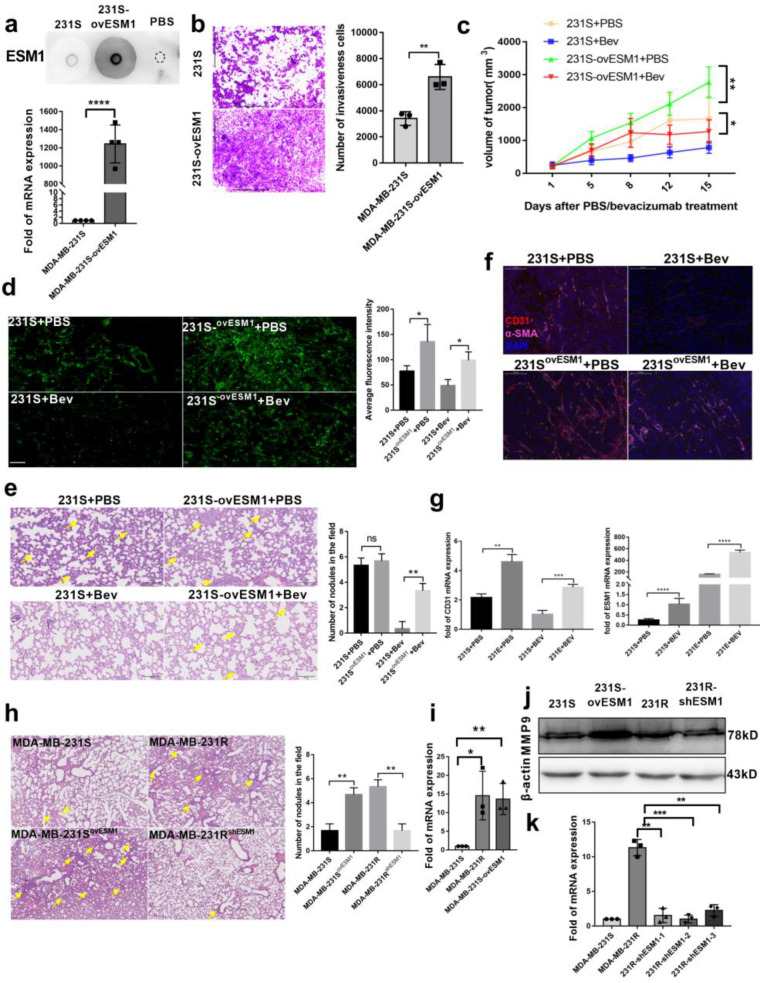
ESM1 promoted bevacizumab-resistance tumor metastasis. (**a**) Dot blotting and qRT-PCR of ESM1 levels in MDA-MB-231-S tumor cells with or without overexpressed ESM1. **** *p* < 0.0001, by *t* test. (**b**) Representative transwell invasion images of MDA-MB-231-S and MDA-MB-231-S ^ovESM1^ cells, ** *p* < 0.01; *n* = 3/group. Scale bar, 200 μm. (**c**) Tumor growth volume over time following subcutaneous transplant of MDA-MB-231-S or MDA-MB-231-S^ovESM1^ tumor tissues into nude balb mice. *n* = 5 for each condition. * *p* < 0.05; ** *p* < 0.01 by two-way ANOVA test. Error bars represent SD. (**d**) Representative images and statistical analysis of MDA-MB-231-S or MDA-MB-231-S ^ovESM1^ lung micro-metastasis treated with PBS or bevacizumab. Scale bar, 100 μm, * *p* < 0.05 by *t* test. (**e**) Representative images and statistical analysis of MDA-MB-231-S or MDA-MB-231-S ^ovESM1^ lung macro-metastasis treated with PBS or bevacizumab. Scale bar, 100 μm, ** *p* < 0.01 by *t* test. (**f**) Representative images of MDA-MB-231-S or MDA-MB-231-S ^ovESM1^ tumor sections from mice treated with PBS or bevacizumab. CD31 was co-stained with α-SMA and nucleus. Scale bar, 50 μm. (**g**) qRT-PCR of CD31 and ESM1 mRNA levels in MDA-MB-231-S and MDA-MB-231-S ^ovESM1^ formed tumor tissue treated with PBS or bevacizumab in vivo. *n* = 3/group. **** *p* < 0.0001, by *t* test. (**h**) Representative images and statistical analysis of HE staining to assess the number of lung metastasis in the different treatment groups; scale, 100μm, ** *p* < 0.01 by *t* test. (**i**) qRT-PCR of MMP9 mRNA levels in MDA-MB-231-S, MDA-MB-231-R, MDA-MB-231-S ^ovESM1^ cells. *n* = 3/group. * *p* < 0.05; ** *p* < 0.01, by *t* test. (**j**,**k**) Western blot and qRT-PCR analysis MMP9 in MDA-MB-231-S, MDA-MB-231-R, MDA-MB-231-S ^ovESM1^ and MDA-MB-231-R ^shESM1^ cells. ** *p* < 0.01; *** *p* < 0.001, by *t* test (Appendix A).

**Figure 4 cancers-14-05681-f004:**
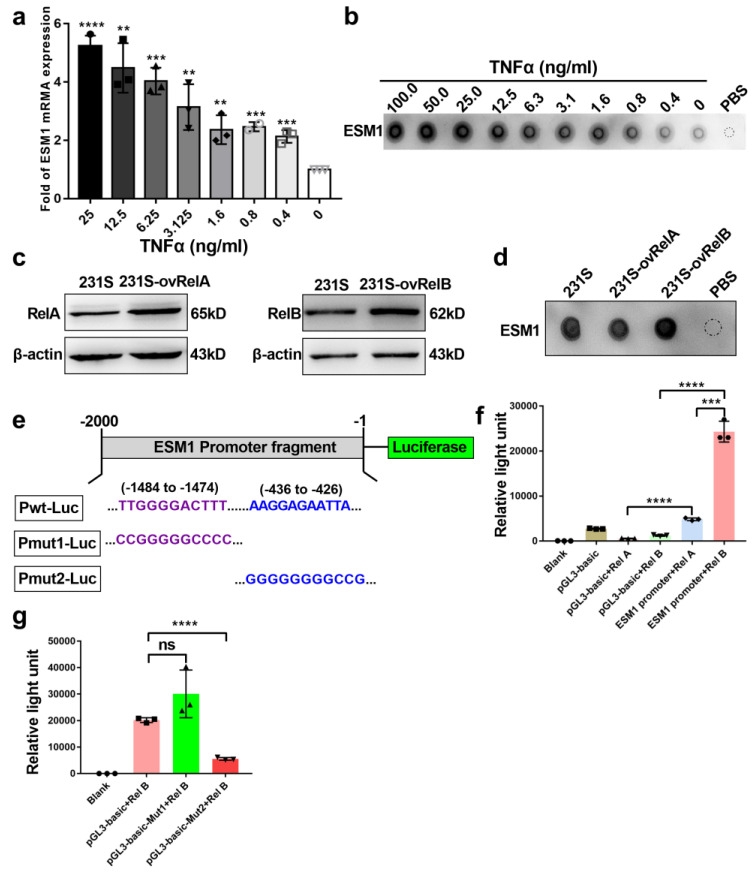
ESM1 is regulated by TNFα through RelB subunit of NF-κB. (**a**) qRT-PCR of ESM1 levels in MDA-MB-231-S tumor cells treated with various concentrations of TNFα (ng/mL). 0 ng/mL TNFα treated group as control, ** *p* < 0.01; *** *p* < 0.001, by *t* test. (**b**) Dot blotting of ESM1 in MDA-MB-231-S tumor cells supernatant under various concentrations of TNFα (ng/mL) treatment. (**c**) RelA or RelB was overexpressed in MDA-MB-231-S cells and detected by western blotting (Appendix A). (**d**) Dot blotting of ESM1 in MDA-MB-231-S tumor cells supernatant after overexpressed RelA or RelB, dot of PBS as control. (**e**–**g**) Design of luciferase reporter assays in HEK293T cells transfected with the indicated plasmids for 24 h, luciferase activity was determined, *** *p* < 0.001; **** *p* < 0.0001, by *t* test.

**Figure 5 cancers-14-05681-f005:**
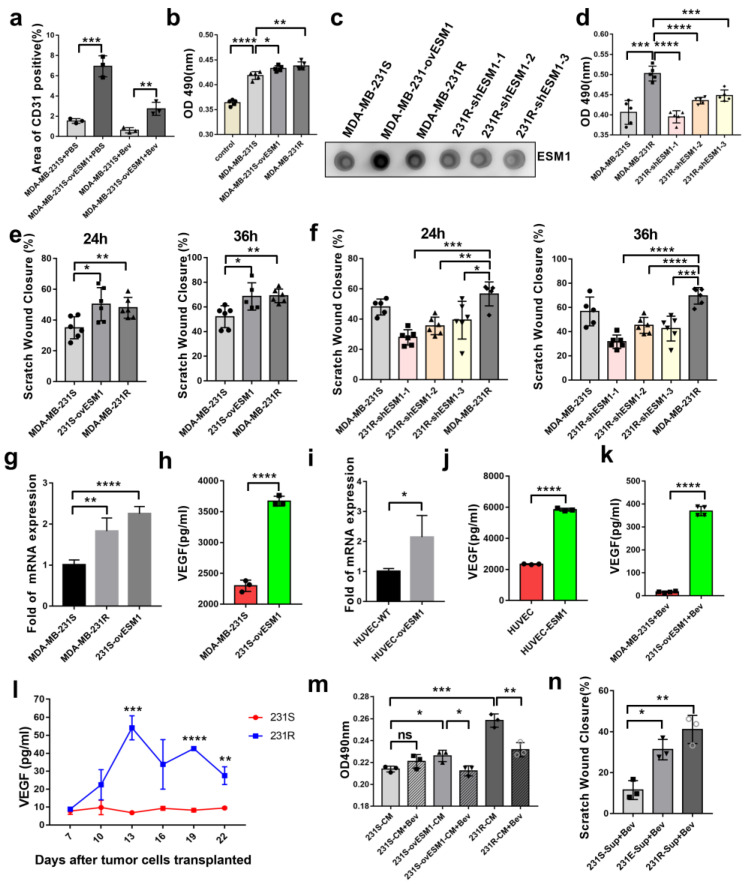
ESM1 promoted the proliferation as well as increased migration of HUVEC in vitro. (**a**) Immunofluorescence analysis of MDA-MB-231-S or MDA-MB-231-S ^ovESM1^ tumor sections from mice treated with PBS or bevacizumab. *, *p* < 0.05; ** *p* < 0.01; *** *p* < 0.001, **** *p* < 0.0001, by *t* test. (**b**) HUVEC cells were incubated by conditioned medium of MDA-MB-231-S, MDA-MB-231-S ^ovESM1^ or MDA-MB-231-R for 24 h and the survival was measured by MTT assay, * *p* < 0.05; ** *p* < 0.01; *** *p* < 0.001, by *t* test. (**c**) Dot blotting of ESM1 in MDA-MB-231-S, MDA-MB-231-S ^ovESM1^, MDA-MB-231-R or MDA-MB-231-R^shESM1#1–3^ tumor cells supernatant. (**d**) HUVEC cells were incubated by conditioned medium of MDA-MB-231-S, MDA-MB-231-R or MDA-MB-231-R^shESM1#1–3^ for 24 h and the survival was measured by MTT assay, *** *p* < 0.001, **** *p* < 0.0001 by *t* test. (**e,f**) HUVEC cells were incubated by conditioned medium of MDA-MB-231-S, MDA-MB-231-S ^ovESM1^, MDA-MB-231-R or MDA-MB-231-R^shESM1#1–3^ for 24 or 36 h and the migration was measured by wound healing assay, * *p* < 0.05; ** *p* < 0.01; *** *p* < 0.001; **** *p* < 0.0001, by *t* test. (**g**) qRT-PCR of VEGFA mRNA levels in MDA-MB-231-S, MDA-MB-231-R, MDA-MB-231-S^ovESM1^ cells. *n* =3/group. (**h**) ELISA for analyzing the level of secreted VEGFA in MDA-MB-231-S and MDA-MB-231-S^ovESM1^ cells. *n* =3. **** *p* < 0.0001 by *t* test. (**i**) qRT-PCR of VEGFA mRNA levels in HUVEC with or without human ESM1 added. * *p* < 0.05, *n* = 3/group. (**j**) ELISA for analyzing the level of secreted VEGFA in HUVEC cells with or without human ESM1 added. **** *p* < 0.0001 by *t* test. (**k**) Serum concentration of VEGFA in MDA-MB-231-S and MDA-MB-231-S ^ovESM1^ bearing mice treated with bevacizumab was detected by ELISA. *n* = 4/group. **** *p* < 0.0001 by *t* test. (**l**) Serum concentration-time curve of VEGFA in MDA-MB-231-S/R bearing mice treated with bevacizumab was detected by ELISA. *n* = 3/group. ** *p* < 0.01; *** *p* < 0.001; **** *p* < 0.0001, by *t* test. (**m**) HUVEC cells were incubated by conditioned medium of MDA-MB-231-S, MDA-MB-231-S ^ovESM1^ or MDA-MB-231-R with or without bevacizumab (0.1 mg/mL) for 24 h and the survival was measured by MTT assay, * *p* < 0.05; ** *p* < 0.01; *** *p* < 0.001, by *t* test. (**n**) HUVEC cells were incubated by conditioned medium that added bevacizumab (0.1 mg/mL) of MDA-MB-231-S, MDA-MB-231-S ^ovESM1^ or MDA-MB-231-R for 24 and the migration was measured by wound healing assay, * *p* < 0.05; ** *p* < 0.01, by *t* test.

**Figure 6 cancers-14-05681-f006:**
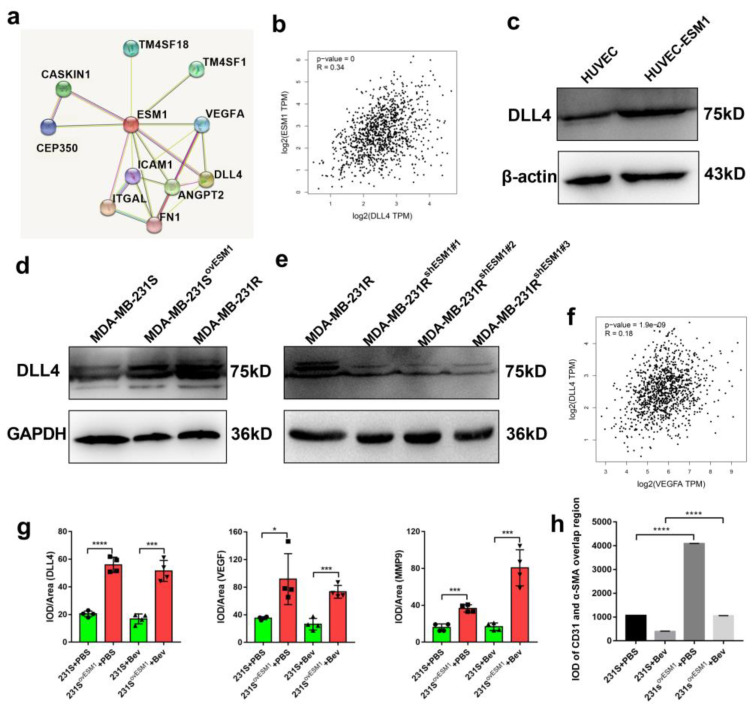
Overexpression of ESM1 increased the expression of DLL4 and promoted productive angiogenesis. (**a**) The protein interaction network was associated with ESM1 in the String Database. (**b**) Gene expression correlation between DLL4 and ESM1 in breast cancer was found in the TCGA database. (**c**) Western blotting for DLL4 in HUVEC cells with or without human ESM1 added (Appendix A). (**d**,**e**) Western blotting for DLL4 in MDA-MB-231-S, MDA-MB-231-S ^ovESM1^, MDA-MB-231-R, or MDA-MB-231-R^shESM1#1–3^ (Appendix A). (**f**) Gene expression correlation between VEGFA and ESM1 in breast cancer was found in TCGA database. (**g**) Quantitative evaluation of DLL4, VEGFA, and MMP9 in MDA-MB-231-S or MDA-MB-231-S ^ovESM1^ tumor sections from mice treated with PBS or bevacizumab. * *p* < 0.05; *** *p* < 0.001 by *t* test. (**h**) IOD of CD31 and α-SMA overlap region in MDA-MB-231-S or MDA-MB-231-S ^ovESM1^ tumor sections from mice treated with PBS or bevacizumab. **** *p* < 0.0001 by *t* test.

**Figure 7 cancers-14-05681-f007:**
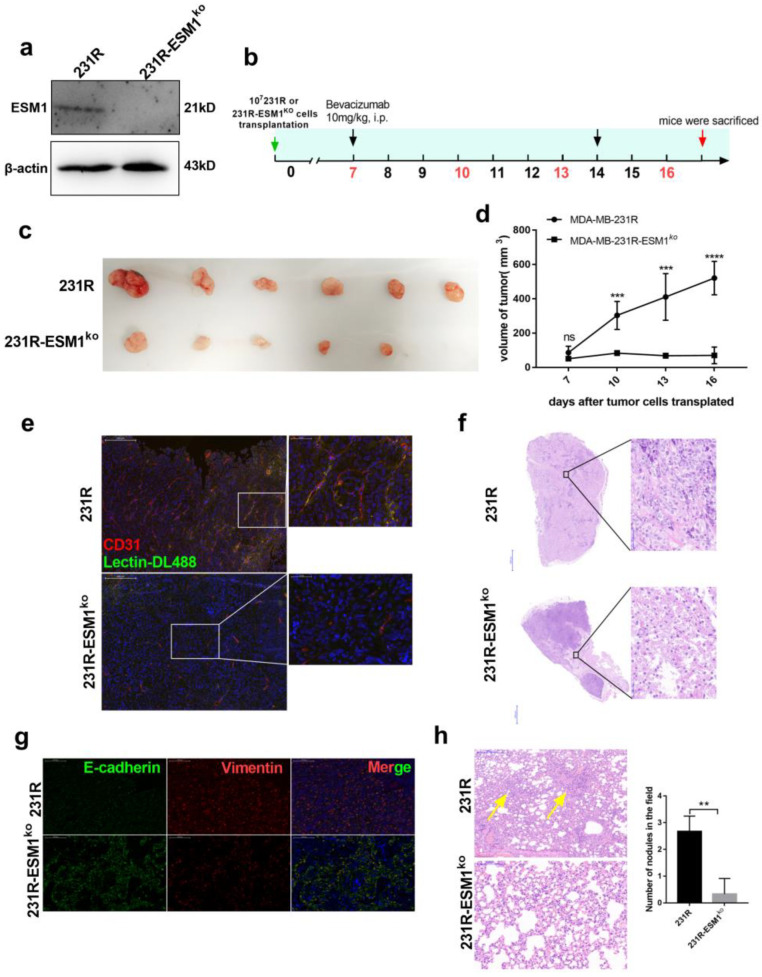
ESM1 deletion in bevacizumab resistant cells and in vivo experiment. (**a**) Western blotting for analyzing the expression of ESM1 deleted or not by the Crispr-Cas9 system in MDA-MB-231-R cells (Appendix A). (**b**) Schematic of experimental design for MDA-MB-231-R or MDA-MB-231-R-ESM1^ko^-bearing mice model treated with bevacizumab. (**c**,**d**) Tumor growth volume and photos over time following subcutaneous transplant of MDA-MB-231-R or MDA-MB-231-R-ESM1^ko^ cells into balb/c-nude mice. *** *p* < 0.001; **** *p* < 0.0001 by two-way ANOVA test. (**e**) Representative images of tumor sections from mice treated with bevacizumab. CD31 was co-stained with DL488-labeled Lectin and nucleus. Scale bar, 200 μm (left) and 50 μm (right). (**f**) Representative H&E staining image of tumor sections from mice treated with bevacizumab. Scale bar, 2000 μm (left) and 50 μm (right). (**g**) Representative images of tumor sections from MDA-MB-231-R or MDA-MB-231-R-ESM1^ko^-bearing mice treated with bevacizumab. Vimentin was co-stained with E-cadherin and nucleus. Scale bar, 200 μm. (**h**) Representative images and statistical analysis of HE staining to assess the number of lung metastasis in the different treatment group; scale bar, 200 μm, ** *p* < 0.01 by *t* test.

**Figure 8 cancers-14-05681-f008:**
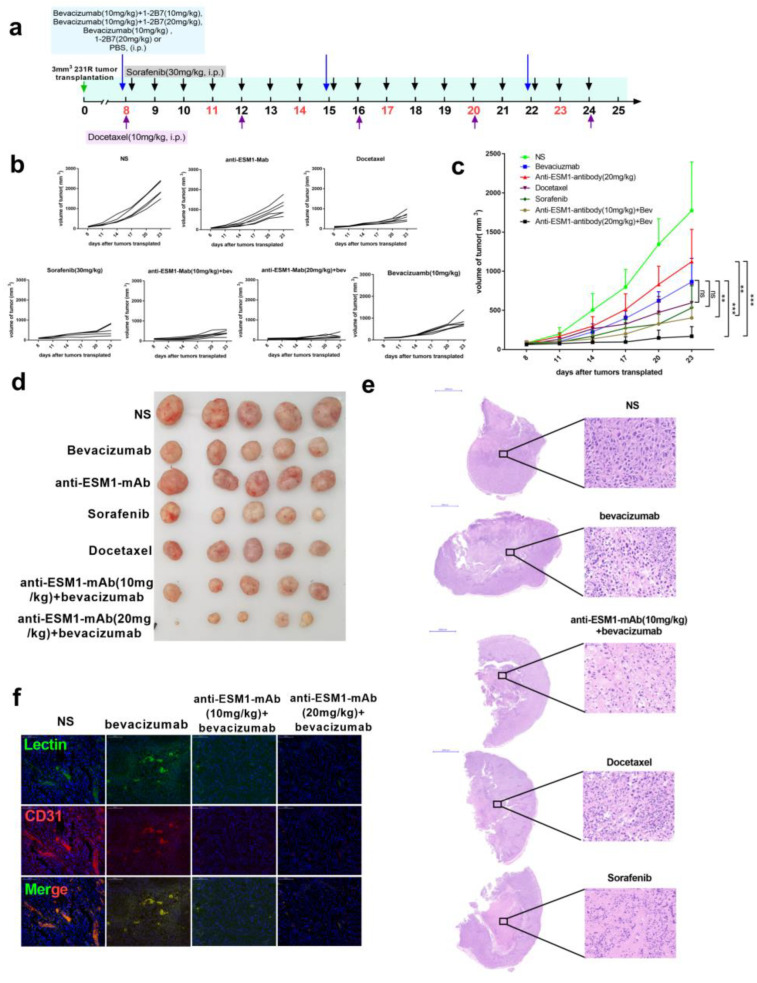
Anti-ESM1 monoclonal antibody alleviated bevacizumab resistance. (**a**) Schematic of experimental design for MDA-MB-231-R-bearing mice model treated with normal saline, docetaxel (10 mg/kg), sorafenib (30 mg/kg) and bevacizumab (10 mg/kg), anti-ESM1-monoclonal antibody (20 mg/kg), or combination bevacizumab with anti-ESM1-smonoclonal antibody with different dosages. (**b**) Tumor growth volume of every mouse in each group over time following subcutaneous transplant of tumor tissues into balb/c nude mice. (**c**,**d**) Tumor growth volume and photos over time following subcutaneous transplant of tumor tissues into balb/c nude mice. ** *p* < 0.01; *** *p* < 0.001 by two-way ANOVA test. (**e**) Representative H&E staining image of tumor sections from mice treated with normal saline, Sorafenib, Docetaxel, bevacizumab and both bevacizumab and anti-ESM1 monoclonal antibody. Scale bar, 2000 μm (left) and 50μm (right). (**f**) Representative images of tumor sections from mice treated with normal saline, bevacizumab, and both bevacizumab and anti-ESM1 monoclonal antibody. CD31 was co-stained with DL488 labeled Lectin and nucleus. Scale bar, 100 μm.

## Data Availability

Data used in this work can be acquired from the String, the TCGA, and the GEPIA database and so on. Further original data inquiries can be directed to the corresponding authors.

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
