# Peer review of "Endothelial-Specific Molecule 1 Inhibition Lessens Productive Angiogenesis and Tumor Metastasis to Overcome Bevacizumab Resistance"

_cancers, 2022, doi:10.3390/cancers14225681_

Round 1

Reviewer 1 Report

In manuscript, Kang et al found inhibiting Endothelial-specific molecule 1 (ESM1) is an efficient way to overcome bevacizumab resistance in cancer both in vitro and in vivo. Moreover, they illustrated the mechanism behind it, showing that inhibition of ESM1 suppress metastasis, angiogenesis through regulating MMP9/ VEGF/ DLL-4 expression. The manuscript is well organized and meaningful to cancer therapy. The following are some major and minor points which need to be addressed.

Major points:

1.     In figure 2f/3a, internal control or groups shown gradient effect need to be added in dot plot for accuracy.

2.     In figure 3d, to show the ratio of GFP positive cells relative to total cells, either DAPI staining or brightfield is needed.

3.     Statistical analysis is encouraged to be performed on images using different slides and different views from each slide.

4.     The quality of fluorescence images is low. Higher resolution is encouraged.

Minor points:

1.     In figure 1e, it looks like DAPI staining is performed in the first row however lost in the second row. Need double check and clarify.

2.     Scale bar is missing in several panels, please double check.

3.     In figure 6b/f, usually a tendency curve will show on graph to indicate positive or negative correlation.

Author Response

We thank you for your thoughtful suggestions and insights, which have enriched the manuscript and produced a better and more balanced account of the research. Furthermore, we would like to show the details as follows:

major points

Point 1:In figure 2f/3a, internal control or groups shown gradient effect need to be added in dot plot for accuracy.

Response 1:  We included the PBS group in the original image of figure 2f. The order of dots from left to right is MDA-MB-231S, MDA-MB-231R, MDA-MB-231R-shESM1#1, MDA-MB-231R-shESM1#2, MDA-MB-231R-shESM1#3, MDA-MB-231R-shESM1#4, and PBS. We did not show the PBS group because the data of the MDA-MB-231R-shESM1s were not required here. To make figure 2f more credible, we reproduced the dot blotting in the supernatant of the same number of MDA-MB-231S and MDA-MB-231R cells. Please refer to the revised figure 2f accordingly. Figure 3a was also updated with the origin image which includes PBS control.

Point 2: In figure 3d, to show the ratio of GFP positive cells relative to total cells, either DAPI staining or brightfield is needed.

Response 2: To analyze ESM1 effect on bevacizumab-resistant tumor metastasis to the lung, we observed the micro-metastasis and metastatic nodules in mouse lungs. The results are displayed in figures 3d and 3e; Figure 3e is the brightfield of Figure 3d.

Point 3: Statistical analysis is encouraged to be performed on images using different slides and different views from each slide.

Response 3: We have added statistical results of the data in the experimental groups to figures 1f, 3d, 3e, 3f (statistical result is 5a), 3h, 7h, where corresponding statistical analyses were required, and the relevant data were added to the figures accordingly.

Point 4: The quality of fluorescence images is low. Higher resolution is encouraged.

Response 4: The resolution of fluorescent images is compressed due to the combination of all images. For the convenience of reviewers and editors to view or use, we have added the original images of immunofluorescence in the attached files accordingly.

minor points

point 1: In figure 1e, it looks like DAPI staining is performed in the first row however lost in the second row. Need double check and clarify.

Response 1: Due to the combination made to the larger picture, the resolution of figure 1e is compressed; therefore, the DAPI staining in the second row may appear to be weakened. We have added the original images so that the fluorescence color of DAPI can be seen clearly in both rows.

Point 2: Scale bar is missing in several panels, please double check.

Response 2: According to your kind reminder, we have carefully checked the scales of all the figures and added them according to the magnification.

Point 3: In figure 6b/f, usually a tendency curve will show on graph to indicate positive or negative correlation.

Response 3: In figures 6b/f, we can infer positive or negative correlations based on R values: positive correlation when R>0 and negative correlation when R<0.

We would like to thank you again for taking the time to review our manuscript.

Reviewer 2 Report

One of the biggest challenges that cancer research is currently facing is the rising proportion of cases that are drug resistant. In this situation, the tumor microenvironment is extremely important. There is a lot of data to support the idea that resistance is linked to alternate pro-angiogenic factor activation. According to research, there is no clinical therapy available to combat such medication resistance, and the majority of patients have developed bevacizumab resistance after receiving continuous treatment. Therefore, bevacizumab resistance can be overcome by having a thorough understanding of the drug's mechanism of action. The authors of this article offer significant insights into the molecular mechanism by which TNF-α induced endothelial cell specific molecule-1 (ESM1) expression encourages angiogenesis. These insights are important for understanding the mechanism of bevacizumab drug resistance and maybe identifying suitable molecules. However, I have few questions.

Why were docetaxel and sorafenib used in mice? How were the mice euthanized?  What was the toxicity parameter following the injection.

What prompted the study of the MDA-MB-231 cell line? In development of the resistant cells, how they make sure the drug resistance is authentic. The authors must include the MDA-MB-231-R cell line evidence. Did authors check for resistance through MDR genes?

What were the results for the MTT assay at different time intervals, such 24 or 48 hours? There is no evidence that 48 hrs. treatments are more effective.

In results 3.1, the authors do not make it clear which mouse flank they are referring to.

To quantify lung metastases, MDA-MB-231-R and MDA-MB-231-S cells were injected into mice through the tail vein, and lung histological sections were observed at 2 and 4 weeks.  Alien cells infiltrated the lungs and formed nodules in the MDA-MB-231-R group (Fig. 1f). Therefore, bevacizumab-resistant MDA-MB-231 cells have increased proliferation and invasion capacity in vitro and in vivo. The whole of this statement is unclear. First, why did they use a tail vein to inject tumor cells? How were lung metastases being monitored? Did authors check Serum Amyloid level A (SAA) for metastasis or through imaging? There is no evidence of metastasis in lung. This entire situation is misleading.

To further confirm that ESM1 could promote tumor lung metastasis, we injected 5 × 105 MDA-MB-231-S, MDA-MB-231-E, MDA-MB-231-R or shESM1-MDA-MB-231-R cells into Balb/c-nu mice via the tail vein. From this statement it is not clear the number of mice used and in what volume the cells were injected.

After treatment, there is no indication of IHC images in the docetaxel and sorafenib group. The authors need to compare the outcomes of those treatments as well.

Author Response

We thank you for your thoughtful suggestions and insights, which have enriched the manuscript and produced a better and more balanced account of the research. Furthermore, we would like to show the details as follows:

Point 1:Why were docetaxel and sorafenib used in mice? How were the mice euthanized? What was the toxicity parameter following the injection.

Response 1: Thank you for your insightful questions and valuable comments and suggestions. First, because docetaxel is the first-line chemotherapeutic drug in triple-negative breast cancer, it presented as the positive control in this experiment. Sorafenib is an anti-angiogenesis drug that inhibits angiogenesis by blocking VEGFR, so it presented as a control group with bevacizumab in the inhibition of angiogenesis study. Second, we wanted to investigate whether the bevacizumab-resistant tumor was sensitive to the chemical medicine or targeted drugs in future research, so we added these two groups to this experiment accordingly.

Further, Mice were euthanized through cervical dislocation. The specific operations are: Mice were first anesthetized. Then, researchers caught the tail root and lifted it on the cover of the mice or other rough surface with the right hand, used the thumb and index finger of the left hand to press down the head and neck of the mice, and grabbed the root of the tail of the mice with the right hand to pull back and up which resulted in cervical dislocation. The spinal cord and brain stem were severed, and the experimental animal died immediately. The toxicity parameter following the injection is weights of the mice. The toxicity parameter following the injection is weights of the mice. We ended our experiments when the weight of any group of balb/c-nude mice was lower than 17 g.

Point 2:What prompted the study of the MDA-MB-231 cell line? In development of the resistant cells, how they make sure the drug resistance is authentic. The authors must include the MDA-MB-231-R cell line evidence. Did authors check for resistance through MDR genes?

Response 2: We chose MDA-MB-231 cell line for research because EMA approved bevacizumab as the treatment for metastatic breast cancer. This study follows on our previous study “Tumor necrosis factor α inhibition overcomes immunosuppressive M2b macrophage-induced bevacizumab resistance in triple-negative breast cancer”(DOI: 10.1038/s41419-020-03161-x). In figure 1 of our previous study, we referred to the characteristics of bevacizumab resistance and the establishment and evaluation methods of the existing bevacizumab resistance model in the clinical study (Reference 1: Eriksson JA, Wanka C, Burger MC, Urban H, Hartel I, von Renesse J, et al. Suppression of oxidative phosphorylation confers resistance against bevacizumab in experimental glioma. J Neurochem 2018, 144(4): 421-430. Reference 2: Jahangiri A, De Lay M, Miller LM, Carbonell WS, Hu YL, Lu K, et al. Gene expression profile identifies tyrosine kinase c-Met as a targetable mediator of antiangiogenic therapy resistance. Clin Cancer Res 2013, 19(7): 1773-1783.). We detected CD31 in drug-resistant tumors after continuous injection of bevacizumab accompanied by three tumor transplantations, which represented microvascular density and VEGF expression level, indicating that the tumor has acquired drug resistance, and MDA-MB-231R is separated from this in vivo drug-resistant model. In figures 1a-d of this study, MDA-MB-231R was consistent with the phenomenon of rapid growth of tumor volume and rebound increase of angiogenesis in clinical studies as proved by our in vivo and in vitro experiments; thus, we concluded that MDA-MB-231R had obtained drug resistance accordingly.

Point 3:What were the results for the MTT assay at different time intervals, such 24 or 48 hours? There is no evidence that 48 hrs. treatments are more effective.

Response 3: We measured the growth curves of MDA-MB-231 cells and HUVEC cells using a Real Time Cellular Analysis (RTCA) before MTT. The growth curve showed that the two types of cells were in logarithmic growth phase for 12-48 h after inoculation with 5000–10000 cells/well; therefore, we have to add MTT within 12-48 h after cell inoculation to analyze cell proliferation. In figure 1a, MTT was added 48 h after MDA-MB-231 cells were inoculated. In figures 5b/d, HUVEC cells were treated with conditional medium for 24 h after 12 h inoculation, and MTT was added for detection. In figure 5m, HUVEC cells were treated with conditional medium for 12 h after 12 h inoculation, and MTT was added for detection. The reason why we added MTT at different time points after inoculation for different experiments was determined according to the cell states, and all MTT addition time points were well within 12-48 hours post inoculation.

Point 4:In results 3.1, the authors do not make it clear which mouse flank they are referring to.

Response 4: In the Results section 3.1, female BALB/c mice (6–8 weeks old) were injected subcutaneously in the left flank with 1×107 MDA-MB-231 cells.

Point 5:To quantify lung metastases, MDA-MB-231-R and MDA-MB-231-S cells were injected into mice through the tail vein, and lung histological sections were observed at 2 and 4 weeks. Alien cells infiltrated the lungs and formed nodules in the MDA-MB-231-R group (Fig. 1f). Therefore, bevacizumab-resistant MDA-MB-231 cells have increased proliferation and invasion capacity in vitro and in vivo. The whole of this statement is unclear. First, why did they use a tail vein to inject tumor cells? How were lung metastases being monitored? Did authors check Serum Amyloid level A (SAA) for metastasis or through imaging? There is no evidence of metastasis in lung. This entire situation is misleading.

Response 5: As demonstrated in figure 1f, we found that the lung colonization ability of MDA-MB-231R cells was stronger than that of MDA-MB-231S cells. However, this did not seem to indicate that MDA-MB-231R was more invasive in vivo. To avoid this misleading discrepancy, we have changed the description “Therefore, bevacizumab-resistant MDA-MB-231 cells have increased proliferation and invasion capacity in vitro and in vivo.” to “Therefore, bevacizumab-resistant MDA-MB-231 cells have increased proliferation and invasion capacity in vitro and the ability to colonize the lungs in vivo.”

Point 6:To further confirm that ESM1 could promote tumor lung metastasis, we injected 5 × 105 MDA-MB-231-S, MDA-MB-231-E, MDA-MB-231-R or shESM1-MDA-MB-231-R cells into Balb/c-nu mice via the tail vein. From this statement it is not clear the number of mice used and in what volume the cells were injected.

Response 6: In this experiment, the number of animals used was n = 3/group, and the injection volume was 100 µL.

Point 7:After treatment, there is no indication of IHC images in the docetaxel and sorafenib group. The authors need to compare the outcomes of those treatments as well.

Response 7: We have added the IHC images of the docetaxel and sorafenib groups accordingly (Please see the attachment).

We would like to thank you again for taking the time to review our manuscript.

Round 2

Reviewer 2 Report

All answers have been provided by the authors. It might be accepted for publication.